# Deciphering the Fate of Burned Trees After a Forest Fire: A Systematic Review Focused on Conifers

**DOI:** 10.3390/biology14101372

**Published:** 2025-10-08

**Authors:** Alessandro Bizzarri, Margherita Paladini, Niccolò Frassinelli, Enrico Marchi, Raffaella Margherita Zampieri, Alessio Giovannelli, Claudia Cocozza

**Affiliations:** 1Department of Agriculture, Food, Environment and Forestry (DAGRI), University of Florence, Piazzale delle Cascine 18, 50144 Florence, Italy; margherita.paladini@unifi.it (M.P.); niccolo.frassinelli@unifi.it (N.F.); enrico.marchi@unifi.it (E.M.); raffaellamargherita.zampieri@unifi.it (R.M.Z.); claudia.cocozza@unifi.it (C.C.); 2Research Institute on Terrestrial Ecosystems (IRET), National Research Council (CNR), Via Madonna del Piano 10, 50019 Sesto Fiorentino, Italy; alessio.giovannelli@cnr.it

**Keywords:** fire regimes, latent tree mortality, fire effects on trees, tree survival assessment

## Abstract

Wildfires are becoming more frequent and intense due to climate change, which threatens forests and makes it harder to understand how trees survive after a fire. Sometimes, trees appear healthy but die months or years later. This latent mortality is difficult to detect and poorly understood, but it affects how forests recover and survive after fires. In this study, we reviewed scientific research focusing on conifer trees’ responses to wild and prescribed fires. We included studies based on observations in natural forests to understand the causes and effects of latent mortality. The research shows that delayed tree death results from a combination of fire damage and additional stresses like drought and pathogen attacks, which affect trees’ ability to transport water and carbon. Our review highlights that more research and monitoring are needed to predict tree survival after fire.

## 1. Introduction

Forests provide a wide range of societal, ecological, and economic services [1]. Fire has been a natural component in forest ecosystems for centuries, affecting their structure and function. However, at present wildfires can directly impact the full supply of these ecosystem services to the community due to the changes in fire regimes in several environments. Changes in social–economical conditions, population growth, migratory flows (e.g., from rural areas to towns or their surroundings) and, last but not least, warming climate is increasing the occurrence and likelihood of wildfire, resulting in severe environmental impacts in fire-prone ecosystems [2] and further limiting the survival probabilities of scorched trees [3]. Unprecedented fire seasons on a global scale raise a series of scientific concerns, highlighting the exigency to comprehend plant community responses to alterations in fire frequency, severity and seasonality [4]. Millions of hectares of forest are affected by annual wildfires, generating complex ecosystem-level responses [5]. Forests are one of the ecosystems that have maintained a constant wildfire trend since the first decade of the 21st century [6]. Furthermore, 23% of global deforestation is driven by wildfires [7]. Fire significantly influences essential processes, such as biodiversity dynamics, nutrient cycling, and carbon fluxes [8]. Additionally, fire regulates forest structure and supports the natural regeneration of fire-adapted species [5]. Yet, fire can act as powerful destructive forces, altering forest structure, composition and developmental trajectories, reducing resource availability and productivity; while negatively impacting ecosystem services [8,9]. In this context, tree mortality has ecological importance. As a consequence, a better understanding of fire influence on forest resilience, along with the ability to predict the survival of trees, remains a primary research aim [5].

The fire regime in forests dynamically influences ecological processes, such as carbon sinks in the most significant terrestrial forest biomes. Wildfires affect the carbon cycle through direct greenhouse gas emission from biomass combustion, the production and redistribution of pyrogenic carbon, the reduction of carbon uptake due to tree mortality and canopy loss, the impaired sequestration capacity of surviving vegetation, and the acceleration of decomposition processes. Additionally, fire can have long-term implications for the carbon balance by affecting the growth of vegetation and the seedlings recruitment [10,11]. Therefore, to minimize potential economic and environmental losses, the systematic monitoring of both endogenous (physiological growth and development) and exogenous (increased temperature and drought, pests) fire stressors represents an important component to take into account for forest management [12]. However, post-fire forest management actions frequently employ burn severity maps to assess areas with different levels of risk and vulnerability, and restoration actions are then defined based on these fire effects. This approach tends to underestimate fire severity and tree damage score due to their limitations in detecting latent tree mortality [13]. Furthermore, many studies adopt varying methods to define, as well as identify the latent mortality; consequently, the complete absence of a standard definition and detection of “latent mortality” may lead to poor comparability of results. An empirical method related to the likelihood of tree vitality and recovery was followed. Scorched trees showing different levels of burnt crown were assigned to a specific class of damage severity. In a codified procedure, trees were either felled or left standing [14]. Furthermore, avoiding felling of scorched trees that show recovery potential can contribute to preserving soil stability, biodiversity, while minimizing adverse impacts on post-fire regeneration [14]. Although salvage logging is commonly applied to reduce economic losses and mitigate the risk of subsequent disturbances after natural events such as fires, windstorms, and insect infestations, its ability to mitigate or increase such risk remains uncertain [15]. The rescue of scorched trees able to survive after a wildfire could be important for selecting seed-bearing trees and orienting natural regeneration towards a forest composition that is more resilient also limiting the spread of alien species. Moreover, this has a detrimental effect on the relationship between final seedling density and initial density, as well as final seedling density relative to total viable seed dispersal [16]. Therefore, a precise estimation of tree latent mortality after a wildfire is essential to develop post-fire land management strategies [17]. Post-fire death can occur either immediately or latent mortality, because of heat-induced injuries and tissue damage affecting the crown, stem, and root systems [18]. These stressors can disrupt the tree’s carbon and water balance, leading to hydraulic failure and carbon starvation, significantly altering non-structural carbohydrate (NSC) dynamics, and ultimately compromising the tree survival [19,20]. This phenomenon, referred to as latent mortality, is further exacerbated by secondary biotic and abiotic stressors, including bark beetles, pathogen activity, interspecific competition, phenological shifts, and climate variability [17]. The latent tree mortality is frequently investigated through remote sensing and survival models to assess the physiological response to fire and to address forest management practices. In this contest, there is an increasing tendency to use a variety of modeling approaches to assess post-fire burn severity and survival rates. The models utilize spatial datasets and satellite indicators to determine fuel, weather conditions, and forest health [3,21]. Moreover, among forest management practices, prescribed fire is widely used to reduce fire hazards by removing live fuels and contributing to ecosystem restoration and resilience [22,23].

Latent tree mortality remains a poorly understood phenomenon. The main gaps concern the lack of methods for detecting and characterizing latent tree mortality at landscape and regional scales, which results in the exclusion of its potential extent, magnitude, and ecological effects [13]. The process is inherently difficult to predict, adding further uncertainty to fire outcomes [24]. For instance, temporal changes in post-fire forest composition and structure are not deeply documented, although considered crucial for assessing fire impacts [25]. Moreover, a better understanding of the physiological response to fire is fundamental for considering the occurrence of wildfire due to climate change, and it is necessary for forest management practices [5]. In the context of reforestation and post-fire ecosystem restoration actions, it could be useful to comprehend the vitality of scorched trees, as they serve as potential conduits for seed dispersal, act as attractors and provide refuge for fauna. Furthermore, it is fundamental to assessing ecosystem impacts, carbon dynamics, as well as developing forest management guidelines. For forest stakeholders, particularly foresters, in-depth knowledge of the fate of scorched trees is useful for providing important management information, especially in light of the limitations that other techniques, such as visual assessment have highlighted. It is evident that reliance on visual assessment as the primary method for determining the fate of trees in the post-fire context results in a substantial underestimation of total mortality. This approach fails to acknowledge the intricate physiological mechanisms and environmental interactions that govern the medium- and long-term survival of trees [26]. In addition, a common problem for technicians is how to correlate the degree of injury produced by wildfire with the probability that the burned trees can recover and grow in the future. In this context, the Italian Ministry of University and Research financed the DIVAS project, titled “Developing innovative methods to assess tree vitality after a wildfire through analyses of cambium sugars metabolism”, addressed to develop biosensors for quickly discriminate recovering and compromised trees. Such tools would provide forest stakeholders with valuable decision-support, helping avoid premature felling, preserve biodiversity and regeneration capacity, and ultimately plan interventions where mortality risk is high. By overcoming the limitations of subjective visual assessment, these innovative approaches have the potential to enhance the accuracy and effectiveness of post-fire management strategies [27]. Nevertheless, despite these advances, important knowledge gaps remain. Although the effect of wildfires on tree vitality has been extensively studied, the mechanisms driving the recovery of scorched trees after exposure to lethal temperatures remain largely unknown. In this contest, a systematic review was conducted to study the current knowledge on the physiological behavior of the tree after a forest fire in depth. The review was aimed at collecting all the information needed to understand plant physiological responses to fire, to support future studies, particularly to support the identification of latent tree mortality. In detail, the review considers conifer species, due to their capacity to tolerate fire, their ecological dominance in numerous fire-prone ecosystems, and the availability of previous studies on their post-fire responses.

## 2. Materials and Methods

A systematic review was conducted by collecting scientific publications from 2000 to 2024. The PICO method guided the literature selection [28]. We established a search strategy answering this main research question: “Deciphering the fate of burned trees after a forest fire: how can we assess the latent mortality of trees?”. According to the upper methodology, the four PICO elements were:Population: every worldwide conifer forest, covered by prescribed fire or wildfire in natural or semi-natural areas during the last twenty-five years (from 31 December 1999 to 31 January 2025;Intervention: every plant’s physiological proxies used to identify any fire-related stress;Control: tree visual assessment related to tree vitality;Outcome: any possible observed physiological outcome.

We conducted the literature research in January 2025 using two different web libraries, Web of Science (WoS) and Scopus, using the following terms, boolean operators to broaden the search, digging deeply within the article title, abstract, and keywords as well:


**Scopus**


((“wildfire”) OR (“prescribed” AND “fire”)) AND ((“forest”) or (“conifer”)) AND ((“latent”) AND (“mortality”)) OR ((“tree”) AND (“mortality”))

**Web of Science** (it does not require the use of quotation marks)

((wildfire) OR (prescribed AND fire)) AND ((forest) or (conifer)) AND ((latent) AND (mortality)) OR ((tree) AND (mortality)).

The results were filtered for the categories Environmental Science, Agricultural and Biological Sciences, Earth and Planetary Sciences, for any document type (Scopus), or Forestry, Ecology, Environmental Sciences, Biodiversity Conservation, Plant Science, Environmental Studies, and Agriculture Multidisciplinary (Web of Science). Duplicate results were removed, and afterwards, we highlighted 895 results for Scopus, 1399 for Web of Science. All studies were selected according to a set of predefined criteria established by our research group. Firstly, we focused exclusively on studies concerning conifer species, as they are particularly relevant in the context of Mediterranean and temperate forest ecosystems. Secondly, in addition to natural wildfires, we also considered prescribed fires as valid environmental stressors, since both disturbances can significantly affect tree physiology and forest dynamics, although under different intensities and management contexts. Thirdly, only studies employing proximal approaches were included, meaning that we excluded remote sensing data in order to focus only on high-resolution, ground-based measurements that directly capture tree-level responses. Finally, studies dealing with plant nurseries, ornamental species or urban forestry context were excluded, and these environments do not reflect the ecological conditions of natural or semi-natural forests and could introduce biases in interpreting stress responses. Consequently, all 2294 results were assessed in a three-step procedure, wherein each stage imposed progressively stricter selection thresholds. To enhance the credibility of the review, we followed the PRISMA method guidelines to enhance the value of the document to the international scientific community.

## 3. Results

### 3.1. Bibliographic Overview

The primary literature screening generated 2294 records in total, with 895 from Scopus and 1399 from Web of Science (WoS). After screening by title, 72 records from Scopus and 62 from WoS were selected for the next step. The selection, based on the abstract level, produced 31 records from Scopus, 16 from WoS. At the full-text screening stage, 9 studies from Scopus and 7 from WoS were ultimately included in the core section of the review (Discussion). In total, 16 studies were collected, representing 0.69% of the entire investigated literature (Figure 1). Even though the relatively small number, such as the rigorous selection criteria, the sixteen articles included represent the actual knowledge on the conifer tree’s latent mortality after a fire-related stress.

### 3.2. Mapping of Study Sites

The final selection included studies from five countries across three continents. Europe was the most represented region (*n* = 7), followed by North America (*n* = 6) and Asia (*n* = 1). The United States contributed the highest number of study cases (*n* = 6), next Italy (*n* = 4), Spain (*n* = 2), and one study each from Sweden and South Korea (Table 1). Based on the World Ecoregion classification [29], the selected studies spanned a wide range of biomes, indicating a significant ecological heterogeneity (Figure 2). The most frequently represented biomes were Mediterranean Forests/Woodlands/Scrub; Temperate Broadleaf and Mixed Forests; Temperate Coniferous Forests. In addition, each of the following biomes were characterized by one study: Boreal Forests and Taiga biomes, Montane Grasslands/Shrublands. According to the Köppen–Geiger climate zone classification [30], the studies were conducted in a wide range of environments. The Mediterranean climate (Csa) was the most represented, with six study sites. The Oceanic/Humid Temperate (Cfb), Humid Subtropical (Cfa), and Cold Semi-arid (BSk) zones each counted two study sites. Finally, the Cold Continental (Dfb, Dwa) and Subarctic/Boreal (Dfc) zones were each represented by a single study. The tree species investigated in the research studies belonged to four genera. The majority of these were *Pinus* species (13), the remaining genera, *Abies*, *Pseudotsuga*, and *Picea*, were represented by single species (Table 2). Thus, the 16 papers selected were defined by 14 research articles, a review, and a technical report (Appendix A).

## 4. Discussion

According to the title of the review “Deciphering the fate of burned trees after a forest fire”, the available literature allows us to assess the latent mortality of trees by considering the effects on physiological processes and the occurrence in different environmental conditions.

### 4.1. Effects of Fire on Tree Vitality and Functioning

The degree of damage to plants affected by fire can vary and can be classified according to the percentage of the epigeal plant’s area affected. However, visual assessment of injury often does not accurately reflect the plant’s physiological condition, meaning lethal internal injuries frequently go unchecked being multifaceted and complex. This section will analyze the physiological traits that occur during a plant fire exposure, focusing on the organ-specific failure, such as the short-term fire-induced damage. Firstly, heat lethal temperatures induce tree death by cambium death, hydraulic failure and/or carbon starvation [32,36,42]. The loss of hydraulic conductivity of the xylem and the depletion of soluble carbon resources and starch in storage compartments are the main process related to the disruption of carbon sources (needles) and sinks (cambium and primary meristems in crown and roots), these issues often result from injuries to roots, cambium, and crowns, which can lead to latent mortality, sometimes occurring months or even years after the fire event [43,44]. Wildfire can cause severe injuries to the tree’s anatomical structure (Figure 3), potentially leading to death or long-term growth and vitality limitations. Hence, the co-occurrence of damage to one or more plant organs can lead to major physiological impairment [36]. Wildfire, as well as the consequent smoldering fire, causes prolonged heating that can directly damage roots within the consumed organic soil horizon (forest floor). In fact, lethal temperatures can penetrate up to 20 cm into mineral soil under smoldering duff conditions [43]. The consequent loss of fine roots, which are abundant in the forest floor [32,43,44], causes an immediate reduction in transpiration rates due to the inability to supply sufficient water to the crown. Hence, this reduction in sap flux is negatively correlated with forest floor consumption [43]. Furthermore, a critical threshold of 30–40% forest floor consumption beneath a tree’s crown is linked to reduced transpiration and, subsequently, tree mortality. Since roots are connected networks of segments, when structural roots near the stem are damaged or completely burned, all roots downstream are also lost [33]. This compromises the tree’s ability to uptake sufficient water and minerals from the soil, leading to stomatal closure, xylem embolism and decline in transpiration [43]. Moreover, root damage also results in the loss of below-ground stored resources, impacting the tree’s ability to repair fire injury (e.g., assimilated carbon may be prioritized for rebuilding damaged fine roots rather than for stem growth) [32]. The vascular cambium, located beneath the bark, is highly susceptible to heat-induced necrosis [33]. Necrosis is generally assumed to occur at temperatures above 60 °C, though prolonged exposure to lower temperatures can also be lethal [32,33]. In addition, when cambium necrosis occurs around the entire stem circumference, the result is girdling, preventing the regeneration of phloem and xylem in that area [42]. This effectively cuts off photosynthate transport to the roots, leading to eventual root reserve depletion and death from water stress [14,36]. Despite all this, some studies suggest that trees may complete seasonal wood formation post-fire, indicating that temperatures were not sufficient to cause cambial death [36,37]. In continuity with cambium, fire can cause necrosis of phloem tissue due to heat conduction through the tree bark [16,36]. Furthermore, phloem disruption impairs the translocation of processed sap, newly assimilated carbon, and other substances from the crown to the rest of the tree (i.e., carbon sinks) [14], leading to immediate or latent tree mortality [14,34]. It is important to assess that glucose in the phloem can serve as a valid indicator of phloem injury; indeed, its absence indicates an increased likelihood of latent tree mortality [14]. Fire also induces a reduction in soluble sugars, resulting in increased vulnerability to cold temperatures, as soluble sugars are essential for frost resistance [34]. In this sense, measuring glucose concentrations in the phloem, cambium, and last xylem rings using a blood glucometer (a portable, inexpensive device) is a novel and user-friendly approach to rapidly assess latent post-fire mortality. Indeed, glucose is a primary energy source, and its concentration can act as a tracer to estimate phloem vitality and detect phloem injury. A glucose differential in the phloem sap has been identified as a valid indicator of phloem injury. This method can help predict the probability of tree death at an early stage, which improves the efficiency and economics of post-fire recovery operations, such as salvage logging [14]. Wildfires can severely impact plant hydraulic function by damaging xylem cell walls, potentially leading to the formation of air bubbles (embolism) or deformation of xylem structures when heat transfer reaches critical levels [32,35]. These structural alterations impair the efficient transport of water from roots to leaves, compromising both hydraulic efficiency and safety [32,35]. Under conditions of low water availability and high evaporative demand, such disruptions can promote embolism formation, accelerating tissue desiccation and cell death. Damage to the architecture of bordered pits may reduce their sealing capacity and increase the risk of hydraulic failure. This type of xylem dysfunction is widely recognized as a primary mechanism contributing to forest decline and tree mortality, often preceding carbon starvation. However, findings in the literature remain inconclusive. While some studies report increased xylem vulnerability to cavitation and structural deformation post-fire [24,34], research on species such as *Pinus pinaster* and *Pinus pinea* indicates no significant changes in hydraulic efficiency or embolism resistance following fire events [34,35]. These contrasting results suggest that the extent of fire-induced xylem damage may depend on species-specific traits and the severity of the fire. Some species may have xylem features that protect them from the critical level of heat exposure that would otherwise cause lasting internal damage [34]. Focusing on a different section of the plant, crown scorch (foliage burnt by heat) is an immediate consequence of fire, indicating a strong injury; hence, it is a reliable predictor of post-fire tree mortality [43]. Moreover, crown injury determines the degree to which carbon acquisition is decreased by fire and the possibility for recovery of foliage from surviving buds [34]. This type of damage is strictly related to the reduction in photosynthetic capacity and stomatal activity, impairing carbon uptake and potentially leading to carbon starvation [16,35,37,38]. However, the impact of crown damage is often more negative when burning occurs during periods of active growth (e.g., spring), due to lower carbohydrate reserves and higher physiological activity [37]. It is notable to add that even with high levels of crown scorch, some trees can survive and recover, especially if apical buds are protected [16,34,37]. Delving deeper into the issues of fire damage in the plant, and their possible detection, nonstructural carbohydrates (NSCs) are key factors for tree functions like growth, metabolism, and defense, such as recovery from disturbance; indeed, they are stored in periods of reduced carbon gain or high demand [34]. The inner bark (secondary phloem) is an important NSCs storage tissue, and NSCs can be preferentially remobilized when carbon supply is reduced. Reed and Hood (2024) [34] studied the NSCs dynamics, underlying that NSCs depletion is partly linked to reduced photosynthetic leaf area, which limits carbohydrate availability. Secondly, the NSCs reduction is a direct consequence of the carbon demand necessary for trees to repair damaged tissues and to regrow foliage. When NSCs concentration falls below critical thresholds, tree recovery capacity decreases, and carbon may be insufficient to support the function, leading to death [33]. Fire survival trees show a swift recovery of NSCs storage, especially if bud damage is minimal and crown scorch is not too high.

### 4.2. Fire Effects in the Environment

It is well known how fire induces direct physiological damage in trees; nonetheless, the phenomenon of tree mortality within forest ecosystems may be attributable to the synergic impact of multiple stressors. These stressors may include drought and insect outbreaks that, combined with fire, aggravate tree injuries. The established relationship between trees and their environment may also be a contributing factor. Furthermore, climate conditions can act as cumulative stressors, aggravating latent mortality over time [16]. As an example, mortality rates in western forests tend to be elevated in high-altitude areas, where trees experience greater environmental stress, and in drier interior regions, where these disturbances happen simultaneously. In fact, tree death occurs when individuals are unable to acquire or transport enough resources to recover from such biotic and abiotic pressures. This process may occur immediately or gradually, depending on exposure to secondary stressors, further compromising tree vitality [44]. In addition, trees that survive fire may become more susceptible to bark beetle infestations (Coleoptera: Curculionidae, Scolytinae) due to reduced defense capacity [39]. Moreover, tree health can be affected by indirect fire effects, including disruptions to carbon and water fluxes, and lead to latent mortality, even in individuals that initially appeared unaffected [32]. Disturbance regimes, forest composition, and fuel loads have often changed in dry mixed conifer forests, leading to increased susceptibility to unusually severe wildfires and pathogen outbreaks. These changes underline the fragility of ecosystems, including those typically considered fire-resistant, and their limited capacity to recover pre-fire forest structures following atypical disturbance events [39]. Fire also plays an important role in forest composition, particularly through the effects of high-severity crown fires that can significantly modify community structures. In southeastern Arizona, such disturbances resulted in a shift from pine-oak forests to more homogeneous stands dominated by oak and newly resprouted hardwoods [39]. In addition to the forest structure, post-fire conditions tend to support the establishment of mesic species, especially when the altered environment is unable to allow the regeneration of xeric ones [26], by changing the forest composition. Despite the preceding discussion, in boreal forests, wildfires are a fundamental ecological process, influencing biodiversity, nutrient dynamics, and energy balance. Moreover, low-severity fire can positively impact tree growth by reducing understory vegetation and decreasing competition for water and nutrients, enhancing carbon fixation and transpiration [32]. The full range of interactions within the plant-environment complex is studied, as reported in the literature, through various plant monitoring tools, including stem increment sensors and sap flow measurement, as well as dendrochronological techniques. These tools assess tree growth dynamics over time and detect changes induced by the cited environmental stressors, thereby providing valuable information at the tree stand level [32]. Dendrochronology allows long-term comparisons of stem growth dynamics [32] for the evaluation of growth rates in crown scorched and uninjured individuals [37], and the assessment of recent and past fire impacts on trees [16]. Furthermore, it enables the identification of post-fire decline and other combined stressors [35]. Dendrochronology is used to reconstruct and assess tree responses across different fire events in the past [35,38], to provide insights into ecophysiological dynamics following disturbances [35] and to predict post-fire mortality [41]. Even though the study requires further investigation, a combination of morphology and site-specific factors, as the most effective predictors of post-fire mortality in Korean red pine (e.g., specifically, bark scorch index, diameter at breast height, such as slope), focusing on simple, non-destructive, and quick methods for land managers was identified [40].

## 5. Conclusions

This review suggests a multidisciplinary approach to encourage future studies regarding tree latent mortality. A combination of field measurements, laboratory analyses, and advanced in situ monitoring techniques might provide technical data to forecast tree mortality. In this context, for instance, the application of prescribed fire, a controlled simulated fire in the field, allows to collect many data regarding fire characteristics (e.g., temperature and duration of heat exposure [45], height/volume [27,43]), effects on tree physiology and growth (such as on sap flux [32,35,43], stem radial growth [32,35], the stable isotopic composition [16,38], concentration of soluble sugars [14,34]) and data on xylem hydraulic damage monitoring [35].

Despite the fire effects on trees being deeply studied, the reviewed literature underlines many research gaps, particularly regarding the tree’s latent mortality. These gaps underscore the need for further, long-term, and technologically advanced research to fully understand the impact of fire and to develop more effective forest management strategies. In particular, studies are often limited by short observation periods, lacking a mechanistic understanding of the physiological processes that lead to latent mortality. Therefore, future research should prioritize long-term monitoring, the integration of multiple stressors, such as drought and pests, and the application of advanced tools (e.g., sap flow sensors, isotopic analyses) to better comprehend the connection between fire damage and tree responses. Moreover, methodological innovations and broader replication across ecosystems are much needed to improve predictions on latent mortality and provide a stronger scientific basis for adaptive management in fire-prone ecosystems.

Firstly, the mechanisms governing tree latent mortality are still unclear and poorly understood. The movement of non-structural carbohydrates in roots after fire and how this influences the tree’s physiological function has not been fully quantified [34]. Secondly, it is not entirely understood how low and moderate severity of fire influences the tree functioning in the long term [32]. In addition, the understanding of the connection between forest fuel reduction treatment and the resulting latent mortality is particularly lacking. The direct and indirect causes of tree mortality, especially for conifers, are still not well defined. The effect of pre-fire tree health on post-fire mortality has been rarely examined or considered in mortality prediction. Studies indicate that factors influencing pre-fire health (drought, competition, pathogens) can partly determine post-fire mortality, especially when we consider latent mortality [45]. Most post-fire mortality studies only include data for one or two years. Whereas, mediums, such as long-term studies on tree post-fire response, could be recommended to improve human knowledge on forest species-specific behavior.

However, this is the first review focused on latent mortality post-fire in conifers. For this reason, the implementation of the research by including broadleaves will be the next aim to reach by supporting the current knowledge for the development of detection tools and application of targeted management techniques post-fire [14]. Although we are aware that the results obtained in this review derive from the application of a single guided method of literature selection, and that other selection procedures, such as VOSviewer, could be applied, we emphasized the importance of using robust methods to select papers for review, thus making the information repeatable and the procedure traceable. Furthermore, different studies define fire intensity differently. Latent mortality is predictably related to fire intensity; therefore, further studies are needed to better understand the link between fire stress, the environment, and the physiological response of trees.

## Figures and Tables

**Figure 1 biology-14-01372-f001:**
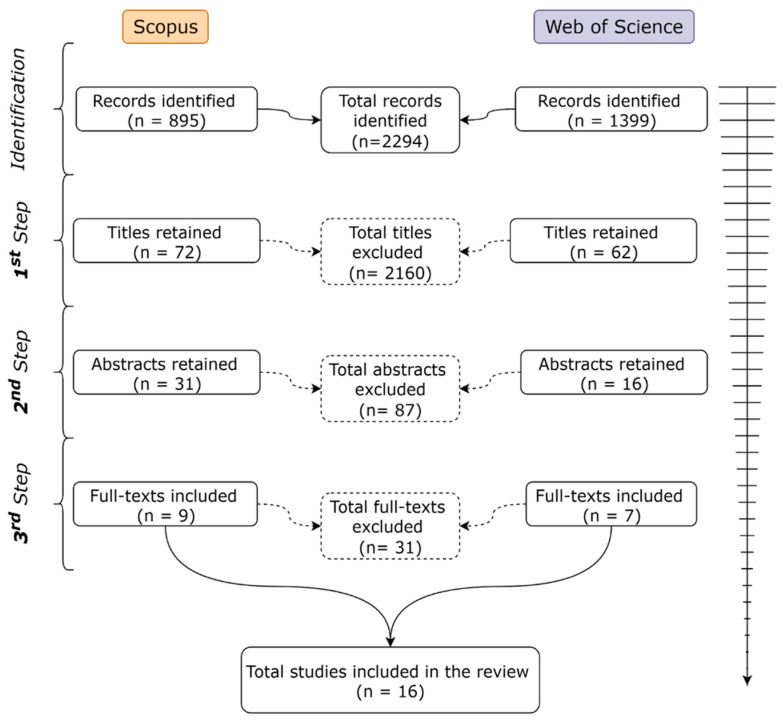
The diagram shows the papers’ selection process. Each field shows the number of selected publications. Dashed boxes represent the excluded articles during the selection process.

**Figure 2 biology-14-01372-f002:**
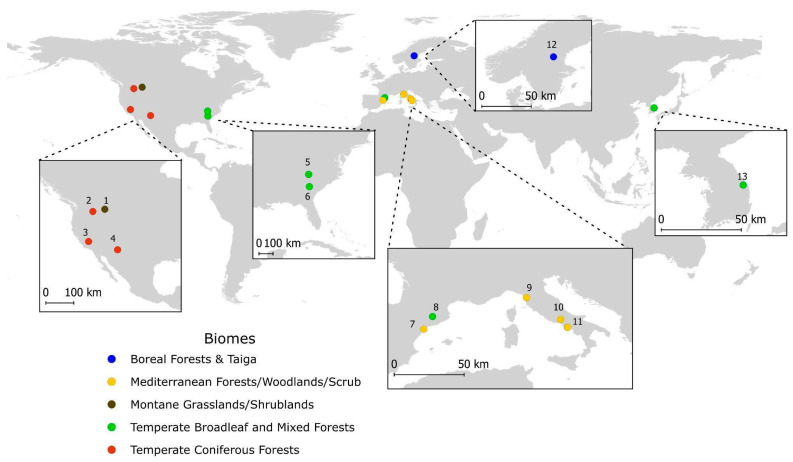
Global distribution of selected papers’ study location and corresponding biomes illustrated as categorized by [31] Each number on the map represent a specific area and the species are distributed as follow: 1, 2: *Pinus ponderosa, Pseudotsuga menziesii;* 3: *Pinus lambertiana, Pinus ponderosa*; 4: *Pinus ponderosa, Pinus strobiformis, Abies concolor, picea engelmanii, Pseudotsuga menziesii;* 5: *Pinus ponderosa, Pinus palustris;* 6: *Pinus virgiliana, Pinus strobus, Pinus taeda;* 7: *Pinus halepensis;* 8: *Pinus nigra salzamannii, Pinus Sylvestris;* 9*: Pinus pinea;* 10*: Pinus nigra;* 11*: Pinus pinaster, Pinus pinea, Pinus halepensis, Pinus nigra;* 12*: Pinus sylvestris;* 13: *Pinus densiflora.*

**Figure 3 biology-14-01372-f003:**
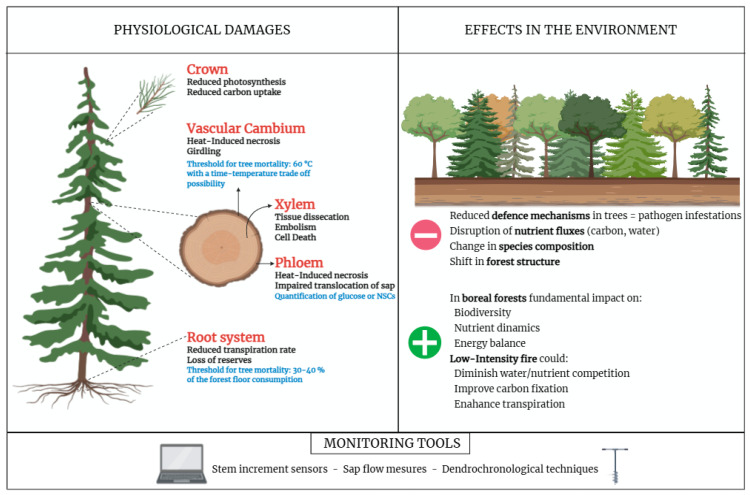
“Physiological damages” and “effects in the environment” highlight the systematic structure of this review by helping readers understand the interconnected, multi-level impacts of latent mortality. On the left, in red, the tissue that sustained damage; in black, the negative impact on the physiology; in blue, the threshold and parameters that should be considered for early detection of latent tree mortality. On the right, the consequences of damage at tree level on the forest environment.

**Table 1 biology-14-01372-t001:** Distribution of the publication across continents and countries.

Continent	Countries Included	N° Studies	References
Europe	Italy, Spain, Sweden	7	[14,16,31,32,33,34,35]
North America	United States	6	[26,36,37,38,39,40]
Asia	South Korea	1	[41]

**Table 2 biology-14-01372-t002:** Number of species across the selected publications.

Genus	N° Species	References
*Pinus*	13	[14,16,26,31,32,33,34,35,36,37,38,39,40,41]
*Abies*	1	[38]
*Pseudotsuga*	1	[37,38,40]
*Picea*	1	[38]

## Data Availability

The data will be made available upon request.

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
