# Peer review of "Deciphering the Fate of Burned Trees After a Forest Fire: A Systematic Review Focused on Conifers"

_biology, 2025, doi:10.3390/biology14101372_

Round 1
Reviewer 1 Report
Comments and Suggestions for Authors
Against the backdrop of accelerating climate change, increasing wildfire frequency, and expanding impact scope, this review systematically explores the critical issue of potential tree mortality following forest fires. It thus holds significant academic value and provides important guidance for post-fire restoration practices.
Deficiencies and Revision Suggestions:
(1) Introduction Section
- Research on restoration ecology against the backdrop of the intensification of global forest fires is an urgent need for humanity. To resonate with readers, it is recommended to add statistical data on the dynamics of global forest fire occurrences to emphasize the severity of the issue.
- Different studies may adopt varying methods to define and detect "potential mortality". The lack of a unified definition and detection standard for "potential mortality" may lead to poor comparability of results. The authors should directly address this issue in the review.
(2) Literature Search
- The author team has conducted a systematic literature search; however, the number of finally included literature is relatively small (16 articles). This may lead to the absence of valuable information, especially in terms of research on the mechanisms of potential mortality, where some important research findings might be omitted.
- Additionally, the inclusion and exclusion criteria for literature were not clearly defined in the selection process, which is critical information to supplement.
(3) Charts and Tables
- The chart information in the review is scattered (e.g., Figure 2, Table 1, Table 2, and Appendices A.1–A.3), requiring readers to cross-reference to understand the correlation among "study area–climate–tree species". It is suggested to integrate the "geographical–climatic–species information" to reduce information fragmentation and intuitively demonstrate the coverage of research samples across global ecosystems.
- Figure 3 (physiological damage) and Figure 4 (environmental impacts) belong to different subsections of the "Discussion" and do not clearly illustrate the causal chain of "tree physiological response → ecosystem changes". Suggestions: Merge Figure 3 and Figure 4, or add arrows/block diagrams in Figure 4 to reference key mechanisms in Figure 3 (such as hydraulic failure, carbon starvation), indicating how they affect ecological processes (e.g., carbon cycle imbalance, hindered species regeneration). This will form a logical closed loop of "individual damage → ecological cascade effects", highlighting the theoretical systematicness of the review and aiding readers in understanding the multi-level impacts of potential mortality.
(4) Discussion
- While the review's title does not specify tree species, the content primarily focuses on conifers (especially Pinus), with insufficient discussion on fire responses in other tree species, limiting the review's applicability. It is recommended to explain this selection bias in the discussion or limitations section.
- Different studies define fire intensity inconsistently, and "potential mortality" should vary under different fire intensities. This important factor should be considered in the discussion.
- Tree mortality caused by fires includes direct and indirect pathways, with distinct mechanisms, and post-fire tree death results from the combined effect of both. The review does not clearly elaborate on the relationship between these mechanisms or derive the necessary monitoring methods for scientifically assessing post-fire "potential mortality". It is suggested to further comb through the literature and construct the discussion section to enhance the review's academic value.
(5) Prospects
- Based on theoretical research and restoration practice needs, the review should clearly identify the main deficiencies of existing studies through systematic literature collation, and propose key focuses for future research, or recommend priority research areas and methodological innovations.
- When mentioning "strengthening long-term monitoring", can appropriate monitoring indicators, frequencies, and durations be specified? If not, what are the possible alternative suggestions?
(6) Others
- Terminology consistency: The paper alternately uses "latent mortality" and "delayed mortality". It is recommended to unify the terminology.
Reviewer 2 Report
Comments and Suggestions for Authors
Forest fires are one of the most common causes of forest transformation and death. This is an issue that is highly relevant all over the world. Nevertheless, general global studies are extremely inadequate. This is particularly relevant when it comes to the significant ecological impact of latent tree mortality. Therefore, the topic of this paper is highly relevant.
The paper makes a good impression. The design of the study is well thought out. The methodology corresponds to the tasks set and is also well thought out. Therefore, there is no doubt about the reliability of the conclusions. I don't have any significant comments.
However, I noticed that the literature reviewed focuses mainly on one tree species (see Table 2). The other three types are presented in only one publication. Therefore, it is necessary to expand the search beyond WoS and Scopus. Perhaps the best results would have been obtained by selecting all publications with DOIs.
I can also recommend expanding the bibliometric analysis. For these purposes, you can use VOSviewer software.
Additional comments:
The authors investigated the latent mortality of trees in conifer forests after wildfires or prescribed burning. Formulation of the question ("what are the mechanisms of latent mortality of trees after a fire") It is well integrated into the international discussion and makes the research potentially relevant for the practice of forest science and fire ecology.
The topic of this paper is highly relevant. This is because the consequences of forest fires are a concern for researchers and policymakers around the world. The authors found and analyzed 2,294 articles published between 2000 and 2024. The study conducted by the authors is well thought out and meticulous, so it closes a gap in the study of the consequences of the hidden death of trees from fires.
The paper adds a thorough synthesis of knowledge about the problem.
The authors chose the method of systematic review using the PICO approach, which is the strength of the study: it provides a structured search and allows you to clearly link the research question with the criteria for selecting publications.
The methodology is described in sufficient detail: time frames (2000-2024), databases (Web of Science and Scopus), search queries, and step-by-step selection of articles are indicated. The quantitative dynamics of the eliminated publications at each stage is also presented. This increases the transparency of the procedure and facilitates reproducibility.
It is advisable to add this section for publication by adding: (1) criteria for evaluating the quality of the included studies; (2) a more detailed explanation of the physiological approaches included in the study; (3) to refine the methodology in accordance with the PRISMA guidelines. These additions will strengthen the credibility of the review's results and enhance its value to the international scientific community.
Bibliographic analysis can be enhanced. For this, I can recommend using VQSviewer.
The use of VOSviewer (Visualization of Similarities Viewer) in a systematic review has several significant advantages that enhance the scientific significance of the work and make the results more transparent to the international community:
VOSviewer allows you to visualize the connections between publications, keywords, authors, and organizations. This helps to identify the main research clusters, dominant topics, and knowledge gaps.
The program builds maps based on the frequency of joint citations or co-authorship, which allows you to understand which scientific areas are most developed and which are just being formed.
The inclusion of maps and network diagrams in a systematic review helps reviewers and readers visually assess that the literature selection was comprehensive and representative.
The conclusions correspond to the presented evidence and arguments, and answer the main question posed.
In general, the tables are informative, and the figures are of acceptable quality.
However, the visualization of the results can be significantly improved when using VOSviewer.
Round 2
Reviewer 2 Report
Comments and Suggestions for Authors
The authors have responded to all my comments. I have no further comments.